# Characterizing Humic Substances from Native Halophyte Soils by Fluorescence Spectroscopy Combined with Parallel Factor Analysis and Canonical Correlation Analysis

**Dongping Liu [1], Huibin Yu [1],\*, Fang Yang [1], Li Liu [1,2], Hongjie Gao [1],\* and Bing Cui [3]**

1 Chinese Research Academy of Environmental Science, Beijing 100012, China; 18339916386@163.com (D.L.); yang.fang@craes.org.cn (F.Y.); Liul215@163.com (L.L.)
2 School of Environment, Liaoning University, Shenyang 110136, China
3 College of Geography and Environment, Shandong Normal University, Jinan 250358, China; cuibing0621srrch@163.com
\* Correspondence: yuhb@craes.org.cn (H.Y.); gaohj@craes.org.cn (H.G.)

**Abstract:** Soil is one of the principal substrates of human life and can serve as a reservoir of water and nutrients. Humic substances, indicators of soil fertility, are dominant in soil organic matter. However, soil degradation has been occurring all over the world, usually by soil salinization. Sustainable soil productivity has become an urgent problem to be solved. In this study, fluorescence excitation-emission matrices integrated with parallel factor analysis (PARAFAC) and canonical correlation analysis (CCA) were applied to characterize the components of fulvic acid (FA) and humic acid (HA) substances extracted from soils from the Liaohe River Delta, China. Along the saline gradient, soil samples with four disparate depths were gathered from four aboriginal halophyte communities, i.e., the Suaeda salsa Community (SSC), Chenopodium album Community (CAC), Phragmites australis Community (PAC), and Artemisia selengensis Community (ASC). Six components (C1 to C6) were identified in the FA and HA substances. The FA dominant fractions accounted for an average of 45.81% of the samples, whereas the HA dominant fractions accounted for an average of 42.72%. Mature levels of the HA fractions were higher than those of the FA fractions, so was the condensation degree, microbial activity, and humification degree of the FA fractions. C1 was associated with the ultraviolet FA, C2 was referred to as visible FA, C3 and C4 were relative to ultraviolet HA, C5 represented microbial humic-like substances (MH), and C6 referred to visible HA. C1, C2, C5 and C6 were latent factors of the FA fractions, determined using the CCA method and could possibly be used to differentiate among the SSC, CAC, PAC and ASC samples. C3, C4, C6 and C5 were latent factors of the HA fractions, which might be able to distinguish the ASC samples from the SSC, CAC and PAC samples. Fluorescence spectroscopy combined with the PARAFAC and CCA is a practical technique that is applied to assess the humic substance content of salinized soils.

**Keywords:** humic substances; EEM-PARAFAC; canonical correlation analysis; native halophyte soils; estuary delta

---

## 1. Introduction

Humic substances (HS), as typical natural supramolecular components, are predominantly stabilized by faint dispersion and originate from biodegradation [1–3]. In mineral soils, the HS can constitute 70–80% of organic carbon [4–6]. They have a relatively high amount of lignin, originating from plant debris and organic materials, which is prone to be accompanied by fairly high humification

degrees [7–9]. Despite representing a small proportion of soil isolate, they have all sorts of pivotal roles in the carbon–nitrogen cycle, availability of nutrients, and mobilization of toxic organic/inorganic substances by biochemical processes [10,11].

In general, the HS are able to be categorized into three groups: fulvic acid (FA), humic acid (HA) and humin [12,13]. HA has long molecular chains, is dark brown in color, and is soluble in an alkaline solution, which can be conducive to maintain its shape, soil structure and soil nutrients [14,15]. The FA is low in molecular weight and soluble in alkaline and acidic solutions, which can promote mineral decomposition and nutrient release [16]. The humin is insoluble in water at any pH, which exhibits recalcitrance to transformations by microorganisms and contributes to the stable carbon pool [17].

In this study, gas chromatography, high-performance liquid chromatography, atomic absorption spectrographic, ultimate and fluorescence analyses were used to characterize the structure, composition and functionalities of organic matter. Compared with the other methods, fluorescence excitation-emission matrices (EEM) were preferentially selected for the analysis of humic substances from different environments due to the advantages of simple operation, less reagent wastage, good repeatability, high measuring accuracy and rapid detection [18–20]. The technique can provide a comprehensive plot of the organic matter characteristics occurring within a selected range of excitation-emission wavelengths by visual inspection [21]. Parallel factor analysis (PARAFAC)—a useful method of multivariate data analysis—decomposes the three-dimensional data of the organic matter into independent fluorescent components, which can be semi-quantitatively related to the organic matter precursor materials [22]. EEM, in combination with PARAFAC, have been applied in the representation of dissolved organic matter (DOM) in a variety of circumstances [23–25].

In this study, EEM combined with the PARAFAC was applied to identify the FA and HA fractions isolated from native halophyte soils in a large estuarine delta. The objectives were (i) to extract the fluorescent components using the PARAFAC, and compare their relative abundance in the FA and HA fractions; (ii) to deduce fluorescence indices (FI) for identifying the sources, and evaluating the mature, condensation and humification levels of the FA and HA fractions, and (iii) to expose correlations among the fluorescent components with FIs using the CCA, and seek latent factors.

## 2. Materials and Methods

### 2.1. Study Area and Sampling Strategy

The Liaohe River Delta is the largest reed coastal wetland in China, and the second-largest reed production base in the world. It is located on the northern Liaodong Bay coast (40°45′–41°10′ N, 121°30′–122°00′ E), in which the estuary interchange of the Liaohe River and the Daliaohe River is located. The delta is an important migratory corridor for waterfowl between East Asia and Australia, In the delta, some endangered species including red-crowned cranes and black-billed gulls breed freely. It covers 1280 $km^2$, with an average annual precipitation of 650 mm. The delta is extremely significant in preventing floods, purifying water, limiting the encroachment of seawater, regulating climate and maintaining biodiversity through biological and wetland systems. In recent years, some of the reed fields have experienced further deterioration due to them being converted into paddy farmlands, which might gradually result in degradation of the wetland functions [26].

Halophytes are adapted to growing in saline conditions, such as those found in salt marshes [27–29]. Four native halophytes in the delta were selected according to their salinity tolerance, i.e., the strongly salinized Suaeda salsa Community (SSC), the salinized Chenopodium album Community (CAC), the moderately salinized Phragmites australis Community (PAC), and the slightly salinized Artemisia selengensis Community (ASC) [26]. The sampling was carried out in July 2015 (Figure 1). At each sampling site, three soil profiles were gathered in a $3 \times 3$ $m^2$ soil plots using a tube sampler. The soil profile was divided into four layers (0–20, 20–40, 40–60 and 60–80 cm) [25,30]. Before a given soil sample was thoroughly mixed and sieved (<2 mm), the roots and debris were completely removed.

The soil samples were hermetically poured into packages, stored in an incubator, and transported to the laboratory as soon as possible.

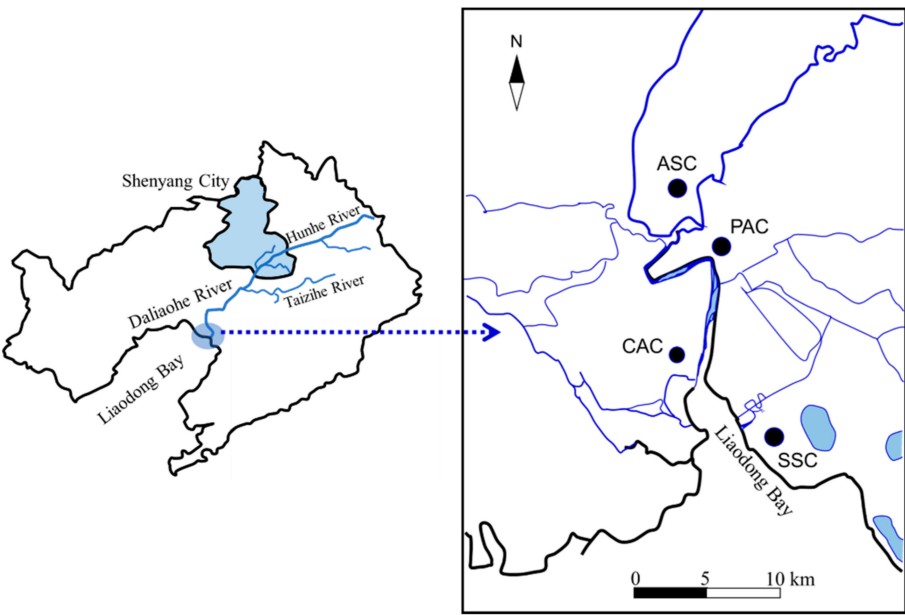

**Figure 1.** The geographical location of the study area.

## 2.2. Physico-Chemical Analyses

A portion of each soil sample was air-dried, crushed and sieved ($\Phi = 100$), while the remaining moist soil samples were analyzed directly. The moist soil samples were analyzed for electric conductivity (EC) and pH, while the air-dried samples were analyzed for the total organic carbon (TOC). The EC was surveyed in the slop (soil/water = 1:1) utilizing an electronic conductivity meter (FE30, Mettler Toledo, Switzerland), and the pH was assayed in the solution (water/soil = 2.5:1) utilizing a pH meter (Sartorius, Germany). A Shimadzu V-CPH analyzer (Japan) was used to measure the TOC [25,26].

## 2.3. Isolation of FA and HA Fractions

The isolation of the HS from the soils was performed according to the methods approved by the International Humic Substances Society [31–33]. In detail, a 20 g air-dried, crushed and sieved sample was added to 100 mL of aqueous solution containing 0.1 M $Na_2P_2O_7$ and 0.1 M NaOH (pH = 13). The suspension was sealed and stirred continuously at 150 rpm for 24 h protected by purging with nitrogen. Then the suspension was centrifuged at 3000 rpm for 15 min. The supernatant was further filtered through the 0.7 μm glass fiber membrane to obtain the FA and HA fractions.

The FA and HA fractions were separated with 0.5 M $H_2SO_4$ aqueous solution. FA and HA appeared in the supernatant and deposition, respectively. The supernatant was centrifuged at 3000 rpm for 15 min to obtain the FA. 100 mL of the 0.1 M NaOH was added in the deposition to obtain the HA. The FA and HA fractions were further filtrated with activated carbon, then the pH of the solutions was adjusted to 8.0 (±0.05) to avoid the precipitation of the HA fraction which might disturb the further analysis [34,35].

## 2.4. EEM Spectroscopy Detection and PARAFAC

A Fluorescence Spectrophotometer (F-7000, Hitachi, Japan) was used to obtain the EEM spectrum. PMT voltage was set at 700 V. Emission (em) wavelengths were conducted from 260 to 550 nm at 5 nm intervals, while excitation (ex) wavelengths were conducted from 240 to 450 nm. The scanning speed was set at 2400 nm·min$^{-1}$ and the response time was 0.5 s. The Rayleigh scattering and the

upper spectral data were defined as 0, and the water Raman scattering was restrained by using Quinine-sulfate [33,34].

The sum of the 64 EEM spectra (the FA = 32 and the HA = 32) was modeled with the PARAFAC by the DOMFluor toolbox 1.7 in MATLAB platform [22]. The number of fluorescence components was determined using split-half and residual analysis. Six principal components (C1 to C6) were extracted from the EEM spectrum, whose maximum fluorescence intensities ($F_{max}$) are representative of the proportional abundances, respectively [22]. The $F_{max}$ of each component was measured as a percentage of the total $F_{max}$ for the six components (%C1 to C6%), which was concerned with the relative abundance.

### 2.5. Multivariable Analyses

The variations of the FA and HA fractions in the different soil profiles were investigated using the matrices of the fluorescent components. The fluorescence indices (Fis)—inversed from the EEM spectra and fluorescent components—were built using the column and Y error plot, and ternary plot. The similarities/dissimilarities of the sampling sites were detected using non-metric multidimensional scaling (nMDS) analysis. The correlations between the fluorescent components and Fis were revealed using bivariate correlation analysis, and the potential factors were sought using canonical correlation analysis (CCA). Statistical analyses were conducted on SPSS 22.0 (SPSS Inc., version 22.0, USA), Origin 8.0 (Microcal, USA) and Canoco 4.5 (Microcomputer Power, USA).

## 3. Results and Discussion

### 3.1. Physico-Chemical Characteristics

The soil EC, pH and TOC were measured in our previous studies on the same samples [26]. A noticeable difference in the TOC can be observed in the different aboriginal halophyte soils along with each soil profile. The average TOC value in the ASC soil profile was 11.36 ± 1.14 mg·kg$^{-1}$, approximately 2.69 times greater than that in the SSC soil profile, 2.04 times greater than that in the CAC soil profile, and 1.49 times greater than that in the PAC soil profile. An evident distinction can be found in the TOC in each soil profile, but there is no similar trend found according to the depth (Figure 2).

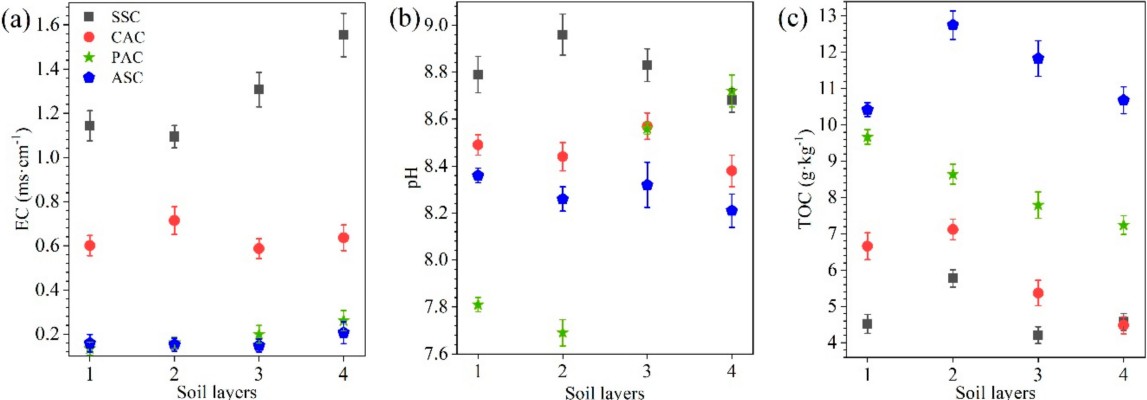

**Figure 2.** Depth-site distributions of physicochemical parameters in the four diverse halophyte communities. (**a**) electric conductivity (EC) (ms·cm$^{-1}$). (**b**) pH. (**c**) total organic carbon (TOC) (g·kg$^{-1}$). Abbreviations: No. 1 = 0–20 cm; No. 2 = 20–40 cm; No. 3 = 40–60 cm; No. 4 = 60–80 cm.

### 3.2. EEM Spectroscopy Characteristics

All the samples of the FA fraction show the similar EEMs—two distinct peaks and one weak shoulder (Figure 3a). Peak A situates within the $\lambda_{Ex}$ from 240 to 280 nm and the $\lambda_{Em}$ from 420 to 480 nm,

could be attributed to the ultraviolet FA substances [35]. Peak C situates at the $\lambda_{Ex}$ of 320–360 nm and the $\lambda_{Em}$ of 400–460 nm, could be ascribed to the visible FA substances [36]. Peak M is situated between the peaks A and C, could be viewed as a microbial humic-like substance (MH), which indicates microbial activity [37]. All of the HA fraction samples show the similar EEM spectra too, in which two strong peaks and one weak shoulder coexist (Figure 3b). Peak D ($\lambda_{Ex/Em}$ = 270–310/450–510 nm) can be ascribed to the ultraviolet HA, and peak H ($\lambda_{Ex/Em}$ = 360–400/450–510 nm) implies the visible HA exists [36,38]. There was a weak shoulder, known as the peak M, which is believed to be MH.

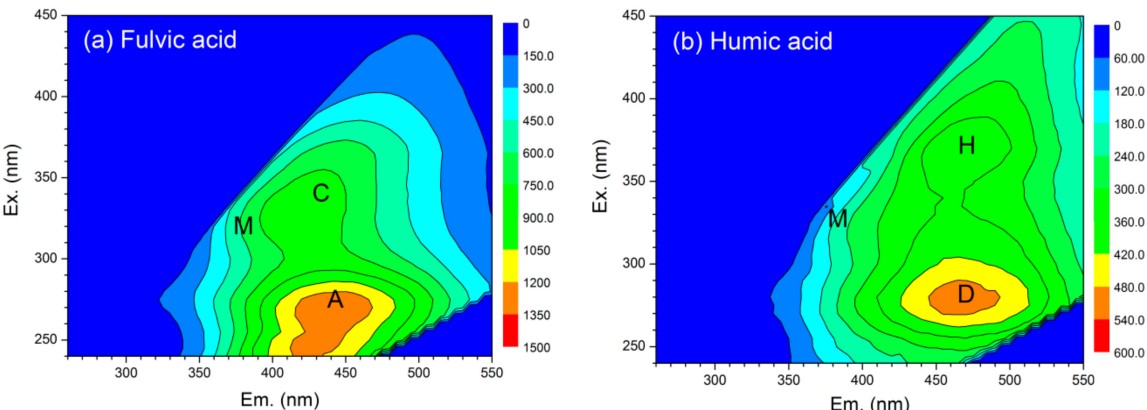

**Figure 3.** The excitation-emission matrices (EEM) spectra of the fulvic acid (FA) (**a**) and humic acid (HA) (**b**) fractions from the Phragmites australis Community (PAC)2 soil sample.

### 3.3. Assignments and Variations of Fluorescent Components

The PARAFAC was applied to analyze the model EEM spectra matrices of the FA and HA fractions and produce six different spectral components (Figure 4). Component one (C1) with the wide and long wavelength peak, resembles the identified peak C, which is associated with the visible FA. Component two (C2) with the wide and short wavelength peak, resembles the identified peak A, which is associated with the ultraviolet FA. The locations of component three (C3) and component four (C4) are similar to the identified peak D, which corresponds to the ultraviolet HA [39]. The location of component six (C6) is similar to the identified peak H, which is relative to the visible HA. Component five (C5) is a ubiquitous fluorescent signal known as the identified M (Figure 4e), which is related to the MH [18].

Concerning the four soil profiles, the average $F_{max}$ of each fluorescent component obtained from the FA fraction is larger than that of the corresponding component of the HA fraction (Figure 5a), especially for C1, whose average $F_{max}$ of the FA fraction (588.56 ± 64.65) is 2.74 times greater than that of the HA fraction. This indicates that the fluorescent components in the FA fraction were much more concentrated than those in the HA fraction, as regards the whole soil profiles. The descending sequence of the mean $F_{max}$ of the components of the FA fraction is C2 > C3 > C1 > C4 > C5 > C6, while the descending sequence of the mean $F_{max}$ of the components of the HA fraction is C2 > C3 > C4 > C1 > C6 > C5. The average %C1 of the FA fraction is higher than that of the HA fraction, as well as the %C2, %C3 and %C5 (Figure 5b). The average %C4 of the FA fraction is lower than that of the HA fraction, similar to the %C6. These indicate that the content of the FA substances in the FA fraction is higher than that in the HF fraction, while the content of the HA substances within the former is lower than that in the latter.

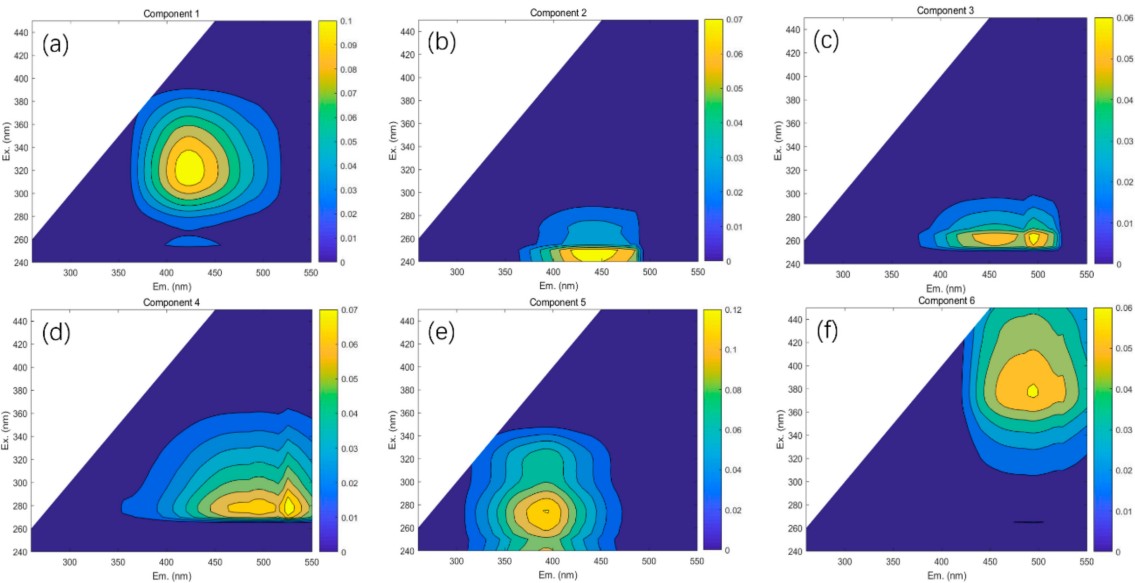

**Figure 4.** The EEM spectrum of the fluorescent components extracted from the FA and HA fractions. (**a**) Visible FA. (**b**) Ultraviolet FA. (**c**) Ultraviolet HA. (**d**) Ultraviolet HA. (**e**) microbial humic-like substance (MH). (**f**) Visible HA.

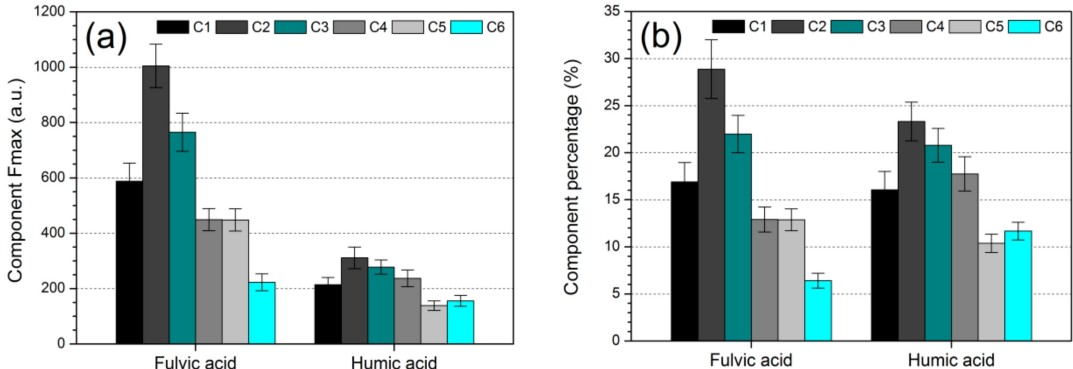

**Figure 5.** The average $F_{max}$ (**a**) and %$F_{max}$ (**b**) of the components in the FA and HA fractions. Three parallel samples were set up and the standard errors across all samples are shown.

For the FA fraction, the mean $F_{max}$ C1 of the ASC soil profile (869.61 ± 163.12) is the highest, followed by the PAC (683.25 ± 163.60), CAC (425.59 ± 130.66) and SSC (375.81 ± 148.44) (Figure 6a). The trends of the average $F_{max}$ C2 and C5 are the same as the average $F_{max}$ C1. The average $F_{max}$ C3 of the ASC soil profile (1017.60 ± 132.57) is the highest, followed by the PAC (786.20 ± 112.01), SSC (641.04 ± 143.09) and CAC (615.42 ± 192.43). The trends of the average $F_{max}$ C4 and C6 are similar to the average $F_{max}$ C3. The variation of each component is distinct in soil profiles, but there is no trend found in the vertical section. Noticeably the $F_{max}$ of each component in the 0–20 cm soil layer is much higher than those in the other soil layers of the SSC soil profile, while the $F_{max}$ of each component in the 40–60 cm soil layer is higher than those in the other soil layers of the PCA soil profile. Further, the $F_{max}$ of a given component in the 60–80 cm soil layer in the CAC soil profile is higher than those in the other soil layers, this was also the case for the ASC soil profile.

With the HA fraction, the descending sequence of the mean $F_{max}$ C1 in the soil profiles is PAC (311.66 ± 261.33) > CAC (261.44 ± 250.11) > SSC (234.15 ± 167.95) > ASC (50.44 ± 29.19), as well as the average $F_{max}$ C2 (Figure 6b). The descending sequence of the mean $F_{max}$ C3 in the soil profiles is SSC (369.16 ± 193.88) > CAC (356.54 ± 274.70) > PAC (333.99 ± 404.89) > ASC (51.58 ± 35.05), as well as the average $F_{max}$ C4 and C6. The descending sequence of the $F_{max}$ C5 in the soil profiles

is PAC (332.37 ± 297.59) > CAC (256.98 ± 246.59) > ASC (171.16 ± 148.12) > SSC (126.90 ± 114.07). The obvious variation of a given component can be found in each soil profile, but there is no trend found in vertical section.

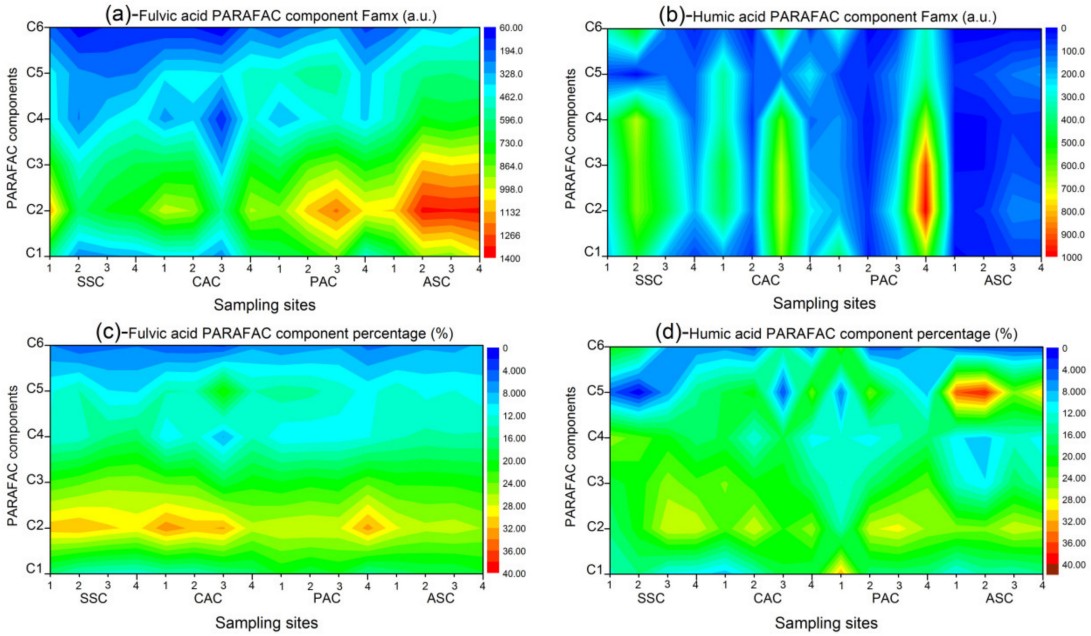

**Figure 6.** Spatial variations of the fluorescent components in the FA and HA fractions. The $F_{max}$ (**a**) and %$F_{max}$ (**c**) of the FA PARAFAC components at all sampling sites. The $F_{max}$ (**b**) and %$F_{max}$ (**d**) of the HA PARAFAC components at all sampling sites.

The percentages of the components varied substantially among the FA and HA fractions (Figure 6c,d). For the FA fraction, the variation range (9.78–22.19%) for the %C5 is the largest, followed by the %C1 (12.4–20.37%), %C4 (7.78–14.62%), %C3 (18.48–25.12%), %C2 (25.90–31.94%) and %C6 (3.39–8.51%). Evidently, the %C2 is higher than those of the other components, suggesting that C2 is the representative component of the FA fraction. As regards to the HA fraction, the variation range (0.53–38.25%) for the %C5 is the largest, followed by the %C1 (8.77–32.01%), %C6 (3.13–23.77%), %C4 (9.07–25.20%), %C3 (11.91–25.05%) and %C2 (14.16–28.85%). Markedly, the variation ranges for the components in the HA fraction are much larger than those in the FA fraction.

The %FA (C1 + C2) in the FA fraction is approximately 45.81%, about 1.12 times greater than the %HA (C3 + C4 + C6), while the %FA in the HA fraction is 39.38%, 1.28 times less than the %HA. The FA is the main component in the FA fraction, and the HA is the major component in the HA fraction. This is consistent with a redshift in the fluorescent maxima of the HA fraction contrast with the FA fraction (Figure 3). The redshift in the fluorescent maxima could be attributed to the existence of high molecular weight fractions, electron-withdrawing substitutes, and a higher conjugation level [1,40]. The %C5 (%MH) is 12.88% in the FA fraction, about 1.24 times higher than that in the HA fraction, indicating that the former had more active microbial sources than the latter.

The variations of the %MH, %FA and %HA in the HA fraction are consistently larger than those for the FA fractions (Figure 7). For the FA fraction, the %FA stays relatively stable (41.17–48.16%) within the whole soil profiles (Figure 7a). The %FA values in the PAC soil profile are nearly high compared with those in the other soil profiles. The %MH remained roughly consistent (9.78–15.19%), except for the CAC3 (22.19%). The variation range for the %HA within the CAC soil profile (29.65–42.38%) is the largest, followed by the SSC (40.56–48.40%), PAC (36.75–40.78%) and ASC (41.40–43.65%). For the HA fraction, the %FA does not change too much (36.23–46.16%) in all soil samples (Figure 7b), except for the SSC1 (29.95%) and CAC1 (31.66%). The %FA values in the PAC soil profile are higher than those in the other soil profiles. The variation range for the %HA in the CAC soil profile (32.32–54.94%) is the

largest, followed by the SSC (45.97–65.45%), PAC (31.83–49.17%) and ASC (22.52–34.34%). The %HA values in the SSC soil profile are higher than those in other soil profiles. The decreasing order of the variation range for the %MH is CAC (3.79–24.42%) > PAC (5.74–24.42%) > SSC (0.53–15.61%) > ASC (24.41–38.25%). The %MH values in the ASC soil profile are the highest contrast with the other soil profiles, which suggests that there might be active microbial sources in the ASC soil profile, rather than in the other soil profiles.

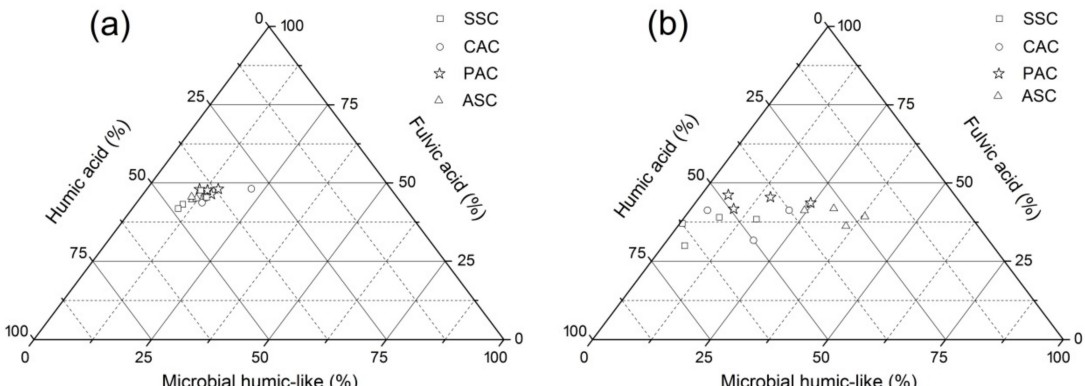

**Figure 7.** Ternary plots illustrating the %MH, %FA and %HA in the FA (**a**) and HA (**b**) fraction for the whole soil samples.

*3.4. Deduction of Fluorescence Indices*

It has been reported that the quotient of the fluorescence intensities at the Ex/Em 370/450 nm and Ex/Em 370/500 nm ($f$450/500) is fluorescent index (FI) which indicates the organic matter sources [41,42]. The $f$450/500 value less than 1.40 indicates the main contribution from allochthonous sources (terrestrial-derived organic matter), larger than 1.90 indicates the main contribution of autochthonous sources (microbial-derived organic matter), and between 1.40 and 1.90 indicates the main contribution of both allochthonous and autochthonous sources, i.e., mixture sources [43,44]. The average $f$450/500 values of the HA fraction are 1.31 ± 0.12 proving that it is mainly derived from the allochthonous sources, while the mean of the FA fraction is 1.49 ± 0.20 proving that it mainly derived from the mixture sources. For the FA fraction, the descending sequence of the mean $f$450/500 values is CAC > SSC > ASC > PAC, and its values stay relatively consistent within each soil profile except for the PAC soil profile (Figure 8a). For the HA fraction, the descending sequence of the mean $f$450/500 is ASC > PAC > CAC > SSC, and its values maintain relatively stable too (Figure 8b). A ratio between the C2 and C1 (peak C: peak A, C/A), is well known as a practical FI to indicate the mature level of the organic matter which is reinforced after a rise in the C/A [1]. The average C/A value of the FA fraction (0.57 ± 0.13) is lower than that of the HA fraction (0.69 ± 0.45), indicating that the mature level of the HA fraction is higher than that of the FA fraction. The decreasing order of the average C/A values in the FA fraction is ASC > PAC > CAC > SSC (Figure 8c), while the decreasing order in the HA fraction is PAC > CAC > SSC > ASC (Figure 8d). A ratio between the C4 and the C3 (C4/C3) is defined as an FI to indicate the condensation degree of organic matter, which increases with the increasing of the C4/C3 [33]. The C4/C3 mean of the FA fraction is 0.58 ± 0.09, approximately 1.44 times less than that of the HA fractions. This suggested that the condensation degree of the FA fraction is lower than that of the HA fraction. The increasing order of the C4/C3 means in the FA fraction is PAC < CAC < SSC <ASC (Figure 8e), while the increasing order in the HA fraction is CAC < PAC < ASC < SSC (Figure 8f).

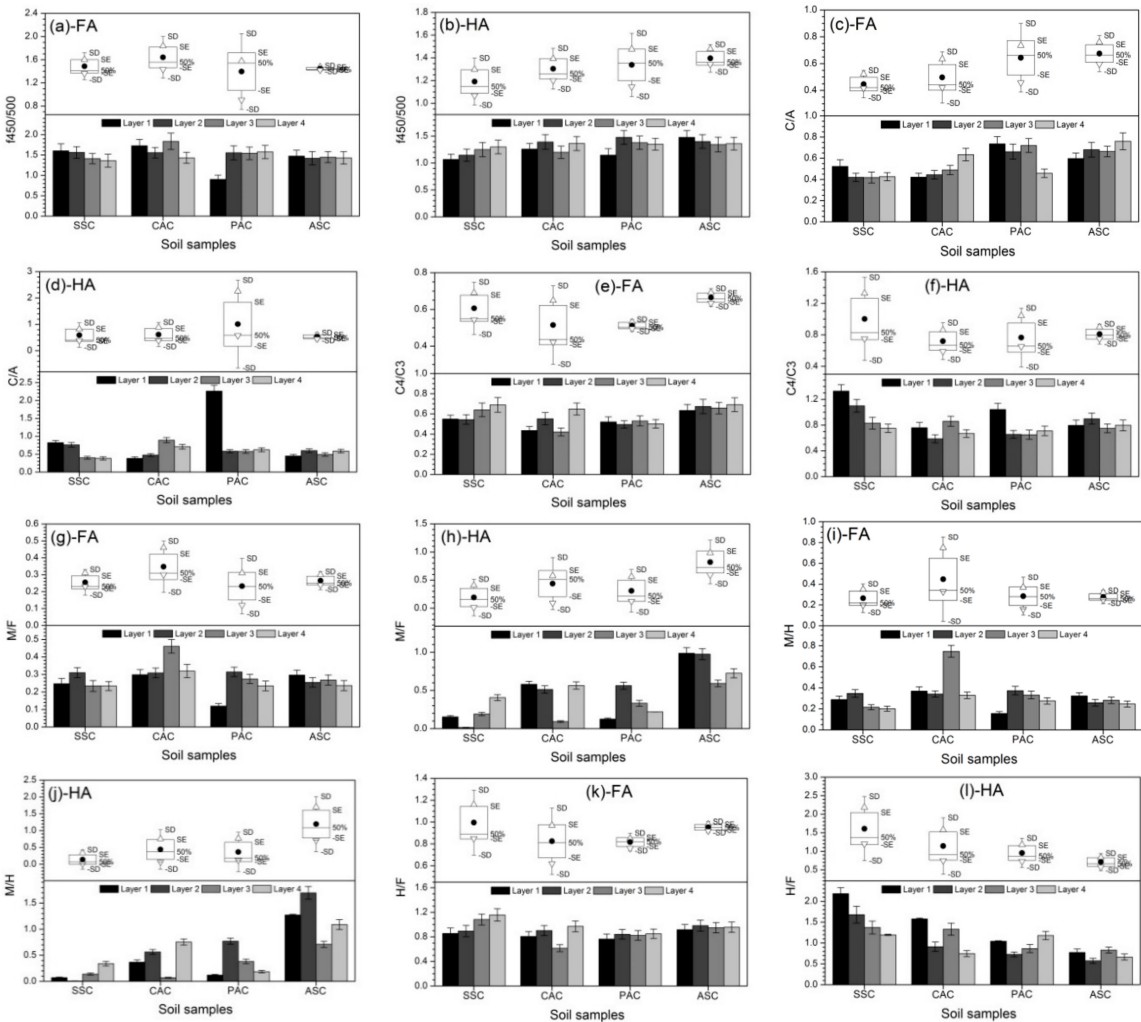

**Figure 8.** Histogram and box-plot symbolize the distributions of the *f*450/500 (**a**,**b**), C/A (**c**,**d**), C4/C3 (**e**,**f**), M/F (**g**,**h**), M/H (**i**,**j**) and H/F (**k**,**l**) of the FA and HA fractions extracted from the soils. Histogram plots show the fluorescence indices (FI) values, black cycles represent the average values, blank triangles present the maximum and minimum values, and whiskers denote standard deviation (SD) values.

A ratio between the C5 and the C1 + C2 (microbial humic-like: fulvic acid, M/F) is well known as an FI to indicate microbial activity, and it increases with a rise in M/F [45]. The M/F mean of the FA fraction is 0.28 ± 0.07, approximately 1.59 times less than that of the HA fraction. This shows that the microbial activity in the FA fraction is lower than that in the HA fraction. For the FA fraction, the decreasing order of the M/F means is CAC > ASC > SSC > PAC, and the M/F variations in the CAC and PAC soil profiles are larger than those within the SSC and ASC soil profiles (Figure 8g). For the HA fraction, the decreasing order of the M/F means is ASC > CAC > SSC > PAC, and there is an obvious variation of the M/F in each soil profile, but no trend can be found in the vertical direction (Figure 8h). A ratio among the C5 with the C3 + C4 + C6 (microbial humic-like: humic acid, M/H), as the M/H, is calculated as an FI to indicate the microbial activity, and it increases with a rise in M/H too [45]. The M/H mean of the FA fraction is 0.32 ± 0.13, approximately 1.68 times less than that of the HA fraction. This identified that the microbial activity in the FA fraction is lower than that in the HA fraction. For the FA and HA fractions, the M/H trends are similar to the M/F (Figure 8i,j). Moreover, a ratio between the C3 + C4 + C6 and C1 + C2 (humic acid: fulvic acid, H/F) has been widely utilized to assess the humification degree of organic matter, and it increases according to an ascent in H/F [35]. The H/F means of the HA fraction (1.10 ± 0.44) is higher than that of the FA fraction (0.90 ± 0.13),

indicating that the humification degree of the HA fraction is high compared with that of the FA fraction. For the FA fraction, the H/F means within the SSC and ASC soil profiles are higher than those in the CAC and PAC soil profiles (Figure 8k). For the HA fraction, the H/F mean in the SSC soil profile is the highest, followed by the CAC, PAC and ASC soil profiles (Figure 8l).

*3.5. Correlations between Fluorescent Components and Fluorescence Indices*

3.5.1. nMDS Analyses

The nMDS was used to analyze the FIs and the fluorescent components of the FA and HA fractions to reveal similarities/dissimilarities between 32 soil samples. The nMDS produces a two-dimensional matrix with stress = 0.13 and $R^2$ = 0.94, suggesting the reliability of the two-dimensional representation [24]. The nMDS map exhibited that the soil samples in the delta are clustered into two groups with the linear equation y = −x, i.e., the FA fraction and HA fraction (Figure 9). This results indirectly show that the FA fraction is significantly different from the HA fraction. The sample desperation of the FA fraction is lower than that of the HA fraction. Therefore, the FA and HA fractions should be analyzed and discussed in a standalone manner.

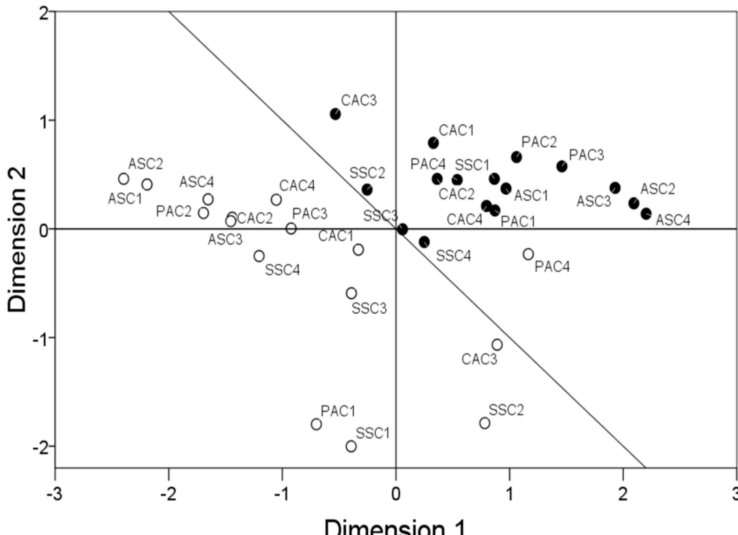

**Figure 9.** Non-metric multidimensional scaling (nMDS) map of the sampling distribution in the delta (blank cycle: the FA fractions and blank cycles: the HA fractions) according to the FIs and fluorescent components.

3.5.2. Correlation Analyses

The $f$450/500 reveals positive relationship with the M/F ($r$ = 0.821, $p$ < 0.01) and MH ($r$ = 0.743, $p$ < 0.01) in the FA fractions, as well as in the HA fractions (MF: $r$ = 0.766, $p$ < 0.01; MH: $r$ = 0.733, $p$ < 0.01). Furthermore, the $f$450/500 presented a better negative relationship with the H/F ($r$ = −0.834, $p$ < 0.01) and C4/C3 ($r$ = −0.797, $p$ < 0.01) in the HA fractions. The M/F shows a positive relationship with the M/H ($r$ = 0.921, $p$ < 0.01) in the FA fractions, as well as in the HA fractions ($r$ = 0.950, $p$ < 0.01). This suggests that the M/F and M/H are practical FIs for indicating the microbial activity. The H/F addresses a negative correlation with the M/H and a positive correlation with the C4/C3 in the FA fractions, as well as the HF fraction. This demonstrated that the humification degree of the humic substances increased with a drop in microbial activity or a rise in condensation degree.

Six FIs in the wind-rose diagrams are clustered into two groups based on the relationships among the FA, HA and MH in the FA and the HA fractions, except for the MH in both fractions (Figure 10). The first group includes the $f$450/500, M/F and M/H, which shows negative correlations with the FA,

HA and MH. The second group contained the C/A, C4/C3 and H/F, which shows positive correlations with the FA, HA and MH. The *r* value between the C/A and FA in the FA fractions is much higher than that in the HA fractions, as well as that between the C/A and HA. This indicates that the C/A is a more feasible strategy to assess the organic matter mature in the FA fractions than in the HA fractions. The FA correlations with the M/F and M/H in the FA fractions are more meaningful than those in the HA fractions, as well as the HA correlations with the M/F and M/H. This implies that the M/F and M/H are more suitable for indicating the microbial activity in the HA fractions than in the FA fractions.

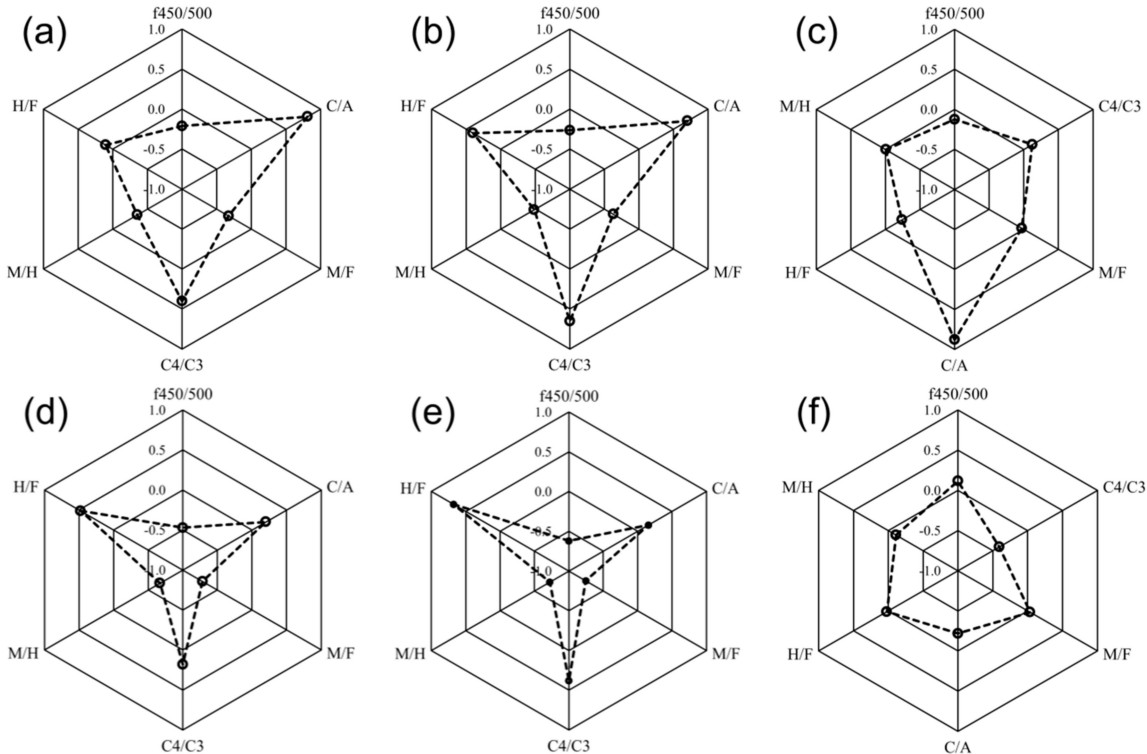

**Figure 10.** Wind-rose diagrams in the FI correlations with the FA (**a**), HA (**b**) and MH (**c**) in the FA fractions, and with the FA (**d**), HA (**e**) and MH (**f**) in the HA fractions.

### 3.5.3. CCA Analyses

The CCA provides an observable and graphic comprehension of the relationships among the FIs, fluorescent components, and samples. Regarding the FA fractions, the *f*450/500, M/F and M/H moved toward the positive direction of the AX1 in the CCA ordination biplot (Figure 11a), while the C/A, C4/C3, H/F and C1 to C6 moved towards the negative direction of the AX1. This indirectly proves that the former showed negative correlations with the latter. The CAC samples remain in the first quadrant, the PAC remains in the second quadrant, the ASC remains in the third quadrant, and the SSC remains in the fourth quadrant. The above results demonstrate that differences in the characteristics of the FA fractions occur among the SSC, CAC, PAC and ASC soil profiles. Based on the law of species and specimens [46], C1, C2, C5 and C6 are the latent factors of the FA fractions, proving that FA is the major component in the FA fractions. The latent factors could differentiate among the SSC, CAC, PAC, ASC samples.

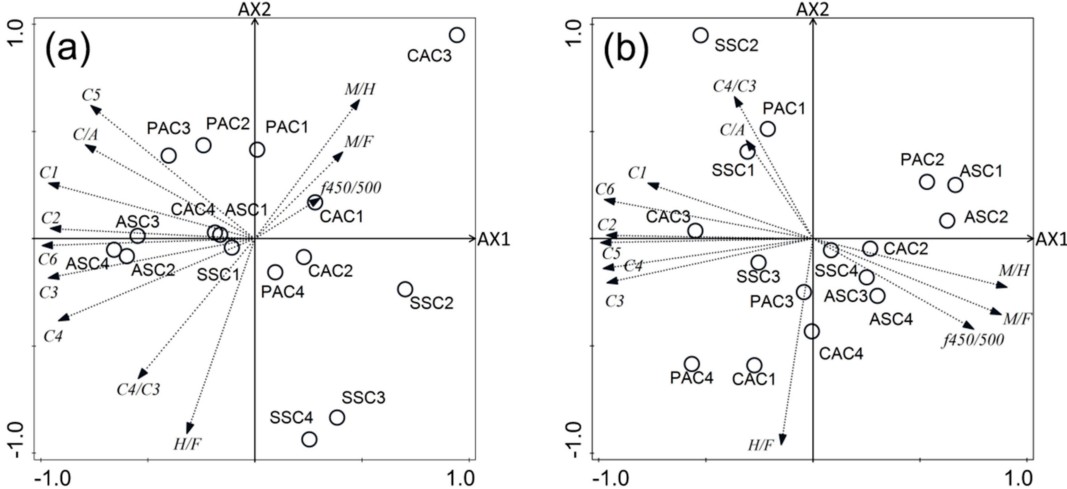

**Figure 11.** The ordination biplot gave canonical correlation analysis (CCA) of the relationships among the FIs, components of the samples in the FA (**a**) and HA (**b**) fractions.

As regards the HA fractions, the arrows of the $f$450/500, M/F and M/H point to the positive direction of the AX1 (Figure 11b), while the arrows of the C/A, C4/C3, H/F and C1 to C6 point to the negative direction of the AX1. This suggests that there should be negative correlations between the former and the latter. The samples in the SSC, CAC and PAC soil profiles (except for the SSC4, CAC2 and PAC2) remain in the second and third quadrants, while the samples within the ASC soil profiles remain in the first and fourth quadrants. This results prove that the characteristics of the HA fractions in the SSC soil profiles are different from the other three soil profiles. Based on the law of species and specimens [46], the C3, C4, C6 and C5 are the latent factors of the HA fractions, proving that HA is the representative component in the HA fraction. The latent factors distinguish the ASC samples from the SSC, CAC and PAC samples.

## 4. Conclusions

The FA and HA fractions extracted for the Liaohe river delta soils are identified as six fluorescent components: C1 is defined as the ultraviolet FA, C2 is the visible FA, C3 and C4 are the ultraviolet HA, C5 is the MH, and C6 is the visible HA. The obvious variations of fluorescent components for the FA and HA fractions are represented in each soil profile, but there is no trend found in the vertical direction. The variations of C1 + C2 (FA), C3 + C4 + C6 (HA), and C5 in the HA fractions are much larger than those in the FA fractions. FA is the dominant component in the FA fractions, whereas HA is the predominant component in the HA fractions. The HA fractions are mainly derived from the allochthonous sources, while the FA fractions are mainly derived from the mixture sources. The mature level of the HA fractions is higher than that of the FA fractions, this was also the case for the condensation degree, microbial activity and humification degree. C1, C2, C5 and C6 are the latent factors of the FA fractions, which are distinct from the SSC, CAC, PAC, ASC samples. The C3, C4, C6 and C5 are the latent factors of the HA fractions, which are different to the ASC samples from the SSC, CAC and PAC samples. The humic substances from soils with different salinized-vegetation improve the buffering potential of the soil and prevent soil from salinization, which might be conducive to the sustainable development of agriculture.

**Author Contributions:** Conceptualization, D.L., H.Y., H.G. and B.C.; data curation, D.L. and H.Y.; methodology, D.L., H.Y. and H.G.; software, D.L., F.Y. and B.C.; formal analysis, D.L. and B.C.; investigation, L.L.; writing—original draft preparation, D.L.; writing—review and editing, D.L., H.Y., F.Y. and H.G.; project administration, H.Y.; funding acquisition, H.Y. and L.L. All authors have read and agreed to the published version of the manuscript.

**Funding:** This research was funded by China Postdoctoral Science Foundation (Fund number, 2013T60148), the Chinese Major Science and Technology Program for Water Pollution Control and Treatment (Grant No. 2018ZX07111001, P.R. China) and the National Natural Science Foundation of China (41907338 and 2019YFC0409205).

**Acknowledgments:** We would like to express our sincere thanks to the anonymous reviewers. Their insightful comments were helpful for improving the manuscript.

**Conflicts of Interest:** The authors declare no conflict of interest.

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
