# Peer review of "Characterizing Humic Substances from Native Halophyte Soils by Fluorescence Spectroscopy Combined with Parallel Factor Analysis and Canonical Correlation Analysis"

_sustainability, doi:10.3390/su12239787_

Round 1
Reviewer 1 Report
The manuscript, titled "Characterizing humic substances from native
halophyte soils by fluorescence spectroscopy combined with parallel factor analysis and canonical correlation analysis " is interesting research, devoted to clarification of nature and origin of the humic substances in overmoisted halpoyte soils of delta of big river, located in Liaohe river delta. This is quite interesting sample plot for detailed chemical investigation of humic substances origin and further transformation. Nevertheless, few suggestions are made with aim to increase the paper quality with major revision decision:
- I suggest to provide picture of insert map of the plots investigated to show the location of sampling plots on the map of China and region
- The soils studied should be described in more details. The pictures of soils and sampling plots should be provided. Soil taxonomy should be given.
- I see that "The soil EC, pH, moisture and TOC were presented and analyzed in previous studies on the same samples", but it is nessesary to give one short table with aim to characterize the rouitine soil properties
- I think that the advantages of Excitation-emission matrix (EEM) fluorescence spectroscopy method should be explained in the introduction chapter more clear. It is necessary to compare this method with other instrumental methods of humiomic science
- it is in contradiction with some previous concepts of humification the statement of the authors that "The HA fractions mainly derived from
the allochthonous sources, while the FA fractions mainly derived from the mixture sources". It is necessary to clarify, what kind of allochtonous sources and what types of mixture sources. - The abstract and conclusion are completely not related to Sustainability topic. Authors have to emphasize the relationship of their material and data to sustainability topic.
Author Response
1. I suggest to provide picture of insert map of the plots investigated to show the location of sampling plots on the map of China and region.
Response/Action: Thanks for the suggestion. We appreciate the suggestion which could help to understand the research profile. Therefore, we tried our best to draw the map of the plots investigated. a regional map of Shenyang with several important rivers was added to the article. Line 90-91
2. The soils studied should be described in more details. The pictures of soils and sampling plots should be provided. Soil taxonomy should be given.
Response/Action:Thanks for the remainder on this point. Following the influence of saline gradient, four native halophytes in the delta were chosen. The soils of which gathered using a tube sampler. According to the soils and sampling plots, we tried our best to collect more related information. Unfortunately, the pictures of soils are not accessible either from the sample collection or published literature. Therefore, we cannot provide the pictures of soils. The sampling plots has been shown in Fig.1 Line 81-91
3. I see that "The soil EC, pH, moisture and TOC were presented and analyzed in previous studies on the same samples", but it is nessesary to give one short table with aim to characterize the rouitine soil properties.
Response/Action:We appreciate the constructive suggestion from the reviewer. In order to make the research results more intuitive, we have added the physicochemical parameters (Fig.2). Line 147-149
4. I think that the advantages of Excitation-emission matrix (EEM) fluorescence spectroscopy method should be explained in the introduction chapter more clear. It is necessary to compare this method with other instrumental methods of humiomic science.
Response/Action:We appreciate the suggestion from the reviewer and we agree that it is necessary to compared Excitation-emission matrix (EEM) fluorescence spectroscopy with other methods of humiomic science. Hence, part of supplementary content was added. “At present, gas chromatography, high-performance liquid chromatography, atomic absorption spectrographic, ultimate and fluorescence analysis were used to characterize the structure, composition and functionalities of organic matter. Compared with the other methods, fluorescence excitation-emission matrices (EEM) were preferentially selected for the analysis of humic substances from different environments due to the advantages of simple operation, small consumption of reagents, good repeatability, high measurement accuracy and rapid detection.” Line 51-56
5. it is in contradiction with some previous concepts of humification the statement of the authors that "The HA fractions mainly derived from the allochthonous sources, while the FA fractions mainly derived from the mixture sources". It is necessary to clarify, what kind of allochthonous sources and what types of mixture sources.
Response/Action:Thanks for the constructive suggestion. The conceptions of allochthonous sources and mixture sources first appeared at line 252-255. “The f450/500 value is less than 1.40 indicating the main contribution of allochthonous sources (terrestrial-derived organic matter), more than 1.90 indicating the main contribution of autochthonous sources (microbial-derived organic matter), and between 1.40 and 1.90 indicating the main contribution of both allochthonous and autochthonous sources, i.e. mixture sources.” It is a quietly mature research findings, and the references have been added in article [25,26]. The fluorescence index (FI) has been used to distinguish the origin of organic substance in natural environments. FI values around 1.9 are characteristic of organic matter derived humic substances (autochthonous origin) whereas terrestrially derived humic substances (allochthonous origin) show FI values around 1.4. Additionally, the values of FI are between 1.4 and 1.9 was considered that organic matter was derived from mixture sources, which contains autochthonous and allochthonous origin. Line 251-255
6. The abstract and conclusion are completely not related to Sustainability topic. Authors have to emphasize the relationship of their material and data to sustainability topic.
Response/Action:Thanks for the suggestion. To emphasize the relationship of our material and data to sustainability topic, we rewrite the abstract and conclusion. The review emphasizes that soil is a renewable resource, while salinization causes soil degradation. The research on humic substances from soils can help alleviate this problem. “Soil is the material basis of human life and the guarantee of agricultural production, which is closely related to human survival and development. However, the soil has degraded all over the world, usually by soil salinization. Sustainable agriculture has become an urgent problem to be solved.” “The study of humic substances from soils with diverse salinized vegetation made an important contribution to improve the buffering power of soil and alleviate soil salinization, which provides significant support for the sustainable development of agriculture.”
Line 13-16;379-382

Reviewer 2 Report
In the present study the authors combine the use of analytical technologies like Excitation-emission matrix with statistical methods like parallel factor analysis to understand the distribution of humic substances in native soils. The results are presented in precise and straightforward manner and I have some minor recommendations before this manuscript could be accepted for publication.
- The abstract has too many jargons and terminologies that makes the purpose of study difficult to understand from just reading the abstract. It is recommended that the authors reword the abstract and explain the purpose of the study in plain and simple english. Words like allochthonous and autochthonous sources etc. can be omitted from the abstract section and can be incorporated in the discussion section.
- Line 47: Change “molecule chains” to “molecular chains”
- Line 48: Change “nutrient” to “nutrients”.
- Line 49: Change “molecule weights” to “molecular weights”.
- Line 59: Change “PARAFAC has being applied” to “PARAFAC has been applied”
- Line 64: Change “deduced” to “deduce”
- Line 82: Explain briefly here, what are “Halophytes” for the understanding of the general reader.
- Line 99-100: The isolation of the HS from the soils was performed according to the methods approved by International Humic Substances Society.
- Line 107-111: “The FA solution of was separated from the precipitation of the HA fraction with the centrifugation at 3,000 r min-1 for 108 15 min. The HA solution of was obtained with the precipitation of the HA fractions added to 100 mL of the 0.1 M NaOH, which was corrected for the volume of 100 mL of 0.1 M Na2P2O7 with the 0.1 M 110 NaOH.”-Very long and unclear sentence. Needs revision.
- Line 112: Change “pH of the solutions should be adjusted to 8.0” to “pH of the solutions was adjusted to 8.0”
- Line 127: Change “%” to “percentage”. Use words and not symbols denoting words.
- Line 128: The word “concerned” seems out of context, do you mean corrected?
- Line 135” Change “busing” to “using”
- Line 137: Change “Canoco 4.5 for windows” to “Canoco 4.5 on windows”
- Figure 2: Show the color scale for all the figures.
- Line 182: Change “which” to “while”
- Figure 3: Denote the number of replicates (n=??) from which this average was calculated.
- Figure 6 description (page 8, line 276): The word “delimit” seems out of context, do you mean denote?
- Line 280: Change “identified” to “denoted”
- Line 319-320: “This indicated that the humification degree of the humic substances increased with a drop in microbial activity or a rise in condensation degree”- Is this observation consistent with other studies described in the literature? If yes, provide references.
- Finally, the conclusion must talk about the significance and broader scope of this study and how these observations can help in better understanding of soil ecology.
Author Response
1. The abstract has too many jargons and terminologies that makes the purpose of study difficult to understand from just reading the abstract. It is recommended that the authors reword the abstract and explain the purpose of the study in plain and simple english. Words like allochthonous and autochthonous sources etc. can be omitted from the abstract section and can be incorporated in the discussion section.
Response/Action:Thanks for the suggestion. To improve the logic of the abstract and explain the purpose of the study in plain and simple english, we rewrite the abctract carefully and many jargons and terminologies has been incorporated in the discussion section. Line 13-33
2. Line 47: Change “molecule chains” to “molecular chains”
Response/Action: Thanks for the careful review. We changed the phrase to “molecular chains”. Line 46-46
3. Line 48: Change “nutrient” to “nutrients”.
Response/Action:Thanks for the careful review. We changed the word to “nutrients”. Line 47-47
4. Line 49: Change “molecule weights” to “molecular weights”.
Response/Action:Thanks for the careful review. We changed the phrase to “molecular weights”. Line 48-48
5. Line 59: Change “PARAFAC has being applied” to “PARAFAC has been applied”
Response/Action:We appreciate the careful revision by the reviewer. We corrected the grammatical problem. Line 61-62
6. Line 64: Change “deduced” to “deduce”
Response/Action:Thanks for the careful review. We changed the word to “deduce” Line67-67
7. Line 82: Explain briefly here, what are “Halophytes” for the understanding of the general reader.Response/Action:We appreciate the suggestion. We explained the halophytes in the materials and methods. “Halophytes are plants that adapted to growing in saline conditions, as in a salt marsh [27,29]”. Line 81-81
8. Line 99-100: The isolation of the HS from the soils was performed according to the methods approved by International Humic Substances Society.
Response/Action:Thanks for the remainder on this point. We revised the sentence accordingly. “The isolation of the HS from the soils was performed according to the methods approved by the International Humic Substances Society”. Line 101-102
9. Line 107-111: “The FA solution of was separated from the precipitation of the HA fraction with the centrifugation at 3,000 r min-1 for 108 15 min. The HA solution of was obtained with the precipitation of the HA fractions added to 100 mL of the 0.1 M NaOH, which was corrected for the volume of 100 mL of 0.1 M Na2P2O7 with the 0.1 M 110 NaOH.”-Very long and unclear sentence. Needs revision.
Response/Action:We appreciate the careful revision by the reviewer. We revised the sentence accordingly. Line 109-111
10. Line 112: Change “pH of the solutions should be adjusted to 8.0” to “pH of the solutions was adjusted to 8.0”
Response/Action: Thanks for the careful review. We changed the sentence to “pH of the solutions was adjusted to 8.0”. Line 112-112
11. Line 127: Change “%” to “percentage”. Use words and not symbols denoting words.
Response/Action:We appreciate the careful revision by the reviewer. We changed “%” to “percentage”. Line 127-127
12. Line 128: The word “concerned” seems out of context, do you mean corrected?
Response/Action:Thanks for the careful review. “The Fmax of each component was measured as a percentage of the total Fmax for the six components (%C1 to C6%), which was concerned with the relative abundance”What I want to express is that the former was related to relative abundance. The meaning of “concerned” here was the same as the related. Line 128-128
13. Line 135” Change “busing” to “using”
Response/Action: Thanks for the careful review. We corrected this error. Line 135-135
14. Line 137: Change “Canoco 4.5 for windows” to “Canoco 4.5 on windows”
Response/Action: Thanks for the careful review. We corrected the grammatical problem. Line 137-137
15. Figure 2: Show the color scale for all the figures.
Response/Action:We appreciate the careful revision by the reviewer. The color scale for all the figures has been added in figure.2. Line 173-175
16. Line 182: Change “which” to “while”
Response/Action:Thanks for the careful review. We changed the word to “while” Line 186-186
17. Figure 3: Denote the number of replicates (n=??) from which this average was calculated.
Response/Action:We appreciate the careful revision by the reviewer. The number of replicates was three, which was mentioned in line 85.”At each sampling site, three soil profiles were gathered in a 3 × 3 m2soil plots using a tube sampler.” During the experiment, three parallel samples were measured. Line 85-85
18. Figure 6 description (page 8, line 276): The word “delimit” seems out of context, do you mean denote?
Response/Action: Thanks for the careful review. We changed the word “delimit” to denote. Line 281-281
19. Line 280: Change “identified” to “denoted”
Response/Action:We appreciate the careful revision by the reviewer. We changed “identified” to “denoted”. Line 285-285
20. Line 319-320: “This indicated that the humification degree of the humic substances increased with a drop in microbial activity or a rise incondensation degree”- Is this observation consistent with other studies described in the literature? If yes, provide references.
Response/Action:Thanks for the constructive suggestion. “This indicated that the humification degree of the humic substances increased with a drop in microbial activity or a rise in condensation degree”, which was obtained by correlation analyses. Through the analysis of correlation coefficient values, we got this conclusion. Additionally, we tried our best to find the references in published literature, unfortunately this kind of message is not accessed. Therefore, we cannot provide the relativereferences. Line324-325
21. Finally, the conclusion must talk about the significance and broader scope of this study and how these observations can help in better understanding of soil ecology.
Response/Action:Thanks for the suggestion. To make the conclusion clear and concise, we deleted detailed results and increasedthesignificance and broader scope of this study. “The study of humic substances from soils with diverse salinized vegetation made an important contribution to improve the buffering power of soil and alleviate soil salinization, which provides significant support for the sustainable development of agriculture.” Line 379-382

Round 2
Reviewer 1 Report
I see that all my suggestions are taken into acocunt, but I reccomend to provide at least soils horizons names in material and methods section and soil names also. This would be response on the key issue of this journal - sustainability of what? - sustainability of soil chemical system. But, every soil have a name.
My desigion is minor revision without correspondnse with me
Author Response
1. I see that all my suggestions are taken into acocunt, but I reccomend to provide at least soils horizons names in material and methods section and soil names also. This would be response on the key issue of this journal - sustainability of what? - sustainability of soil chemical system. But every soil has a name.
Response/Action: Thanks for the constructive suggestion. We appreciate the suggestion which could help to understand the research background. Therefore, we tried our best to supplement relevant content in material and methods. “Four native halophytes in the delta were chosen according to its salinity tolerance, i.e. the strongly salinizedSuaeda salsa Community (SSC), the salinizedChenopodium album Community (CAC), the moderately salinizedPhragmites australis Community (PAC), and the slightly salinized Artemisia selengensis Community (ASC)”. “Within saline soils, strongly salinized soils, salinized soils,moderately salinized soils and slightly salinized soils, the sampling sites were named SSC, CAC, PAC and ASC, respectively”.
According the sustainability, soil is a kind of renewable resources. However, the soil has degraded all over the world, usually by soil salinization. Soil is the material basis of human life and the guarantee of agricultural production, from which humic substances are well known as an indicator of soil fertility. The humic substances from soils with different salinized-vegetation could improve the buffering power of soil and alleviate soil salinization, which provides significant support for the sustainable development of agriculture. Line 81-89; 151-152
